# Effect of PCSK9 inhibition with evolocumab on lipoprotein subfractions in familial dysbetalipoproteinemia (type III hyperlipidemia)

**Elisa Waldmann, Liya Wu[¤], Kristina Busygina, Julia Altenhofer, Kerstin Henze, Alexander Folwaczny, Klaus G. Parhofer** *

Medical Department IV, LMU Klinikum Grosshadern, Munich, Germany

¤ Current address: Department of Gastroenterology, Gastrointestinal Oncology and Endocrinology, University Medical Center Goettingen, Goettingen, Germany

* klaus.parhofer@med.uni-muenchen.de

## Abstract

**Data Availability Statement:** The paper contains the minimal dataset. Individual patient data cannot be provided as patient consent does not include consent for provision of individual level data.

### Background and aims

Familial dysbetalipoproteinemia (FDBL) is a rare inborn lipid disorder characterized by the formation of abnormal triglyceride- and cholesterol-rich lipoproteins (remnant particles). Patients with FDBL have a high risk for atherosclerotic disease. The effect of PCSK9 inhibition on lipoproteins and its subfractions has not been evaluated in FDBL.

### Methods

Three patients (65±7 years, 23±3 kg/m$^2$, 2 females) with FDBL (diagnosed by isoelectrofocusing) and atherosclerosis (coronary and/or cerebro-vascular and/or peripheral arterial disease) resistant or intolerant to statin and fibrate therapy received evolocumab (140mg every 14 days). In addition to a fasting lipid profile (preparative ultracentrifugation), apoB and cholesterol concentrations were determined in 15 lipoprotein-subfractions (density gradient ultracentrifugation; d 1.006–1.21g/ml) before and after 12 weeks of evolocumab treatment. Patients with LDL-hypercholesterolemia (n = 8, 56±8 years, 31±7 kg/m$^2$) and mixed hyperlipidemia (n = 5, 68±12 years, 30±1 kg/m$^2$) also receiving evolocumab for 12 weeks were used for comparison.

### Results

All patients tolerated PCSK9 inhibition well. PCSK9 inhibitors reduced cholesterol (29–37%), non-HDL-cholesterol (36–50%) and apoB (40–52%) in all patient groups including FDBL. In FDBL, PCSK9 inhibition reduced VLDL-cholesterol and the concentration of apoB containing lipoproteins throughout the whole density spectrum (VLDL, IDL, remnants, LDL). Lipoprotein(a) was decreased in all patient groups to a similar extent.

**Funding:** The authors received no specific funding for this work.

**Competing interests:** We have read the journal's policy and the authors of this manuscript have the following competing interests: KGP has received research funding and/or honoraria for consultancy and/or speaker's bureau and/or DMC activity from: Akcea, Amarin, Amgen, Berlin-Chemie, Biomarin, Boehringer-Ingelheim, Dr. Schär, Daiichi-Sankyo, MSD, Novartis, Regeneron, Sanofi, and Silence Therapeutics. This does not alter our adherence to PLOS ONE policies on sharing data and materials. EW, LW, KB, KH, JA and AF have nothing to declare.

**Abbreviations:** FDBL, Familial Dysbetalipoproteinemia; PCSK9, proprotein convertase subtilisin/kexin type 9.

## Conclusions

This indicates that the dominant fraction of apoB-containing lipoproteins is reduced with PCSK9 inhibition, *i.e.* LDL in hypercholesterolemia and mixed hyperlipidemia, and cholesterol-rich VLDL, remnants and LDL in FDBL. PCSK9 inhibition may be a treatment option in patients with FDBL resistant or intolerant to statin and/or fibrate therapy.

## Introduction

Familial dysbetalipoproteinemia (FDBL) is an uncommon form of mixed hyperlipidemia characterized by the accumulation of triglyceride- and cholesterol-rich lipoproteins (remnant particles) in plasma [1]. FDBL is caused by mutations in apoE, typically homozygosity for apoE2, which represents a mutant form of apoE, an apoprotein relevant for the metabolism of triglyceride- and cholesterol-containing lipoproteins [2]. However, homozygosity for apoE2 is not sufficient to induce the characteristic dyslipidemia, dysbetalipoproteinemia (formerly known as type III hyperlipidemia). The development of dysbetalipoproteinemia follows a second hit (environmental factors, other genetic defects, comorbidities). If this secondary factor can be eliminated, lipid profiles may become "normal" again. If dysbetalipoproteinemia is present, it is associated with premature cardiovascular disease [3, 4]. The pathophysiology behind this dyslipidemia relates to the fact that apoB and apoE can bind to the LDL-receptor. Remnants are cleared from the plasma due to the interaction of apoE with the LDL-receptor. ApoE2 has the lowest affinity to the LDL-receptor of all apoE isoforms. This is why in E2/E2 patients remnant particles accumulate in plasma [5]. Although mixed hyperlipidemia (elevated triglycerides and elevated cholesterol) is the typical dyslipidemia in these patients, some may also present with other forms of dyslipidemia such as isolated hypertriglyceridemia or hypercholesterolemia [1].

The clinical management of FDBL is complicated by the fact that the determination of LDL-cholesterol is not reliable as remnant particles may be classified as VLDL or LDL depending on the triglyceride load and method of determination. Therefore, non-HDL-cholesterol is used to guide therapy [1]. Treatment of dysbetalipoproteinemia can be challenging as standard lipid lowering therapies such as statins and fibrates are sometimes not very effective or cannot be tolerated. Niacin, which has been used in the past, is not available anymore in many countries [6]. PCSK9-inhibitors, a new potent treatment option for lowering LDL-cholesterol [7, 8], have not been evaluated in patients with FDBL. PCSK9 is an important regulator of LDL receptor expression but has a number of additional functions as recently described [9]. PCSK9 inhibitors have a strong effect on the lipid profile primarily reducing LDL-cholesterol (-50 to -60%) but also reducing lipoprotein(a) (-20 to -30%). They have little effect on triglyceride levels (-12 to -17%) [10, 11]. Similar to statins and ezetimibe, it was shown that PCSK9-inhibition results in ASCVD risk reduction, and risk reduction is closely related to LDL-cholesterol reduction [7, 8].

For clinical purposes, apoB-containing lipoproteins are usually divided into VLDL and LDL, although all lipoprotein fractions including the LDL fraction represent continua of particles with variable amounts of triglycerides and cholesterol. When gradient ultracentrifugation is used to subdivide the non-VLDL fraction, 15 subfractions are isolated with IDL, LDL and HDL representing subfractions 1–4, 5–11, and 12–15 respectively [12, 13]. Small dense LDL (subfractions 9–11) and probably also large buoyant LDL (subfractions 5–6) are more atherogenic than intermediate dense LDL (subfractions 7–8). Thus, the determination of LDL

subfractions may provide additional information about the atherogenic potential of a lipid profile. Evaluating lipoprotein subfractions is particularly interesting in conditions where the concentration of a broad spectrum of particles is increased, as it is the case in FDBL.

We therefore evaluated the effect of PCSK9-inhibition on lipoprotein subfractions in patients with FDBL and compared it to the effect in patients with LDL-hypercholesterolemia and patients with mixed hyperlipidemia (not FDBL).

## Patients and methods

In this single center study patients with FDBL (apoE2 homozygosity; n = 3), mixed hyperlipidemia (apoE3 or apoE4; n = 5), and isolated LDL-hypercholesterolemia (familial hypercholesterolemia or other forms of isolated hypercholesterolemia; apoE3 or apoE4; n = 8) were recruited from our registry on patients receiving PCSK9-inhibitors. We included all patients with apoE2/E2 phenotype and selected consecutive patients with isolated hypercholesterolemia and mixed hyperlipidemia. All patients had the indication for treatment with PCSK9-inhibitors according to German regulatory standards (not reaching treatment goals recommended by the European guidelines despite maximally tolerated oral lipid-lowering therapy) [14] and received evolocumab 140 mg sc every 2 weeks.

Patients with FDBL were characterized by the typical lipid profile with elevated total cholesterol and elevated triglycerides. In addition, VLDL-cholesterol to VLDL-triglyceride ratio was elevated (>1). All patients suffered from ASCVD, thus either cerebrovascular disease (CVD), coronary artery disease (CAD) or peripheral arterial disease (PAD) and were statin-intolerant with one patient also not tolerating ezetimibe.

Patients with mixed hyperlipidemia all suffered from metabolic syndrome but none from diabetes mellitus. Compared to patients with FDBL, these patients had slightly lower total cholesterol and slightly lower triglyceride concentrations, resulting in an almost identical cholesterol to triglyceride ratio. However in contrast to the patients with FDBL, most of the cholesterol was associated with the LDL fraction and not with the VLDL fraction, resulting in a very different VLDL-cholesterol to VLDL-triglyceride ratio. All patients suffered from ASCVD. Three of the patients did not tolerate statin or ezetimibe, while two were treated with high dose statin and ezetimibe.

Six of the patients with hypercholesterolemia suffered from heterozygous familial hypercholesterolemia (Dutch Lipid Clinic Network (DLCN) Score: probable familial hypercholesterolemia), while 2 did not (DLCN-Score: unlikely or possible familial hypercholesterolemia). All patients had severely elevated total cholesterol because of elevated LDL-cholesterol. All but one patient (patient with heterozygous familial hypercholesterolemia and strong positive family history for atherosclerotic disease) suffered from ASCVD. Four of the patients did not tolerate statins and one did also not tolerate ezetimibe. One patient was treated with additional colesevelam.

The study protocol was approved by the ethics committee of the University of Munich (Protocol number 17–780) and all patients signed informed consent. The study abide by the Declaration of Helsinki principles.

Cholesterol, triglycerides, lipoprotein(a) and apoB concentrations were determined in fasting plasma and different lipoprotein fractions enzymatically before initiation of PCSK9-inhibitor therapy and after 3 months of treatment using an automated clinical chemistry analyser (Response 910, DiaSys, Flacht, Germany). ApoE phenotype was determined once in each subject using isoelectrofocusing as described before [15].

Lipoprotein subfractions were isolated by preparative ultracentrifugation (18 hours, d = 1.006 g/mL, 50000 rpm, 4°C, Beckmann Ti 50.4 rotor). In the supernatant, VLDL-

cholesterol and VLDL-triglycerides were measured. In the infranatant subfractions were isolated by isopycnic density gradient ultracentrifugation, as described previously [12, 13]. Densities were measured by a precision density meter (Anton Paar DMA 38, Graz, Austria). Ultracentrifugation was performed in a Beckmann SW 40 Ti rotor (Palo Alto, CA) at 40 000 rpm for 48 hours at 15˚C. A total of 15 subfractions (SF) were isolated with the following density intervals: SF 1–4, 1.006–1.019; SF 5, 1.020–1.024; SF 6, 1.025–1.029; SF 7, 1.030–1.034; SF 8, 1.035–1.040; SF 9, 1.041–1.047; SF 10, 1.048–1.057; SF 11, 1.058–1.066; SF 12–15, 1.067–1.210 g/mL), with IDL, LDL and HDL representing SF 1–4, 5–11, and 12–15 respectively. Lipoprotein(a) is usually found in SF 11–13; in subjects with elevated lipoprotein(a), the fraction containing the smallest LDL subfraction may therefore contain substantial amounts of cholesterol transported on lipoprotein(a). In all subfractions cholesterol and apoB were determined.

Statistical evaluation: No formal statistical evaluation was performed as group sizes are too small to execute meaningful statistical testing. Descriptive data are given as means ± SD.

## Results

The characteristics of the study participants are shown in Table 1. Most patients received evolocumab because they were either partially or completely statin intolerant. ASCVD was present in all except one subject. Metabolic syndrome was much more common in patients with mixed dyslipidemia. Background lipid lowering therapy was not changed between baseline evaluation and the analysis after 3 months. All subjects tolerated PCSK9 inhibition well and finished the study.

The effects on lipids, apoB and lipoprotein(a) are shown in Table 2, revealing considerable differences between the different patient groups. While the effect on total cholesterol, non-HDL-cholesterol, and apoB is comparable, remarkable differences are seen for triglycerides, LDL-cholesterol and VLDL-cholesterol as well as VLDL-triglycerides indicating a reduction of all apoB containing lipoproteins in FDBL and a predominant reduction of LDL in patients with hypercholesterolemia and mixed hyperlipidemia. In FDBL patients VLDL-cholesterol is reduced much more than VLDL-triglycerides indicating that smaller cholesterol-rich particles are preferentially removed, while the concentration of more typical VLDL is much less affected. In contrast, in patients with hypercholesterolemia and mixed hyperlipidemia, VLDL-cholesterol and VLDL-triglycerides are reduced to a similar extent (Fig 1).

**Table 1. Characteristics of study participants.**

|  | Familial dysbetalipoproteinemia | Mixed hyperlipidemia | Hyper-cholesterolemia |
|---|---|---|---|
| **n** | 3 | 5 | 8 |
| **Male/female (n)** | 1/2 | 3/2 | 2/6 |
| **Age (years) ± SD** | 65 ± 7 | 68 ± 12 | 56 ± 8 |
| **BMI (kg/m$^2$) ± SD** | 23 ± 3 | 30 ± 1 | 31 ± 7 |
| **Metabolic syndrome** | 1 | 5 | 4 |
| **CAD (%)** | 66.7 | 100 | 62.5 |
| **CVD (%)** | 33.3 | 80 | 25 |
| **PAD (%)** | 0 | 20 | 0 |
| **High dose statin therapy (n)** | 0 | 2 | 3 |
| **Any statin therapy (n)** | 0 | 2 | 4 |
| **Ezetimibe therapy (n)** | 2 | 2 | 7 |
| **Other lipid therapy (n)** | 0 | 0 | 1 |

BMI, body mass index; CAD, coronary artery disease; CVD: cerebral-vascular disease; PAD: peripheral arterial disease.

**Table 2. Lipid parameters in different patient populations before and during (12 weeks) of therapy with PCSK9 inhibitors.**

|  | Familial dysbetalipoproteinemia | | | Mixed hyperlipidemia | | | Hypercholesterolemia | | |
| --- | --- | --- | --- | --- | --- | --- | --- | --- | --- |
|  | before | during | (%)[a] | before | during | (%)[a] | before | during | (%)[a] |
| total cholesterol (mmol/l) | 8.64 ± 4.76 | 5.46 ± 2.38 | - 37 | 7.34 ± 1.63 | 4.34 ± 1.45 | - 40 | 6.93 ± 2.38 | 4.94 ± 1.68 | - 29 |
| LDL cholesterol (mmol/l) | 2.02 ± 0.44 | 1.30 ± 0.31 | - 36 | 3.88 ± 1.27 | 1.60 ± 1.14 | - 59 | 4.91 ± 2.07 | 2.95 ± 1.55 | - 40 |
| HDL cholesterol (mmol/l) | 1.24 ± 0.21 | 1.14 ± 0.28 | - 8 | 1.11 ± 0.18 | 1.22 ± 0.23 | + 10 | 1.66 ± 0.80 | 1.58 ± 0.67 | - 4 |
| non-HDL cholesterol (mmol/l) | 7.40 ± 4.94 | 4.29 ± 2.61 | - 42 | 6.23 ± 1.60 | 3.13 ± 1.40 | - 50 | 5.28 ± 2.30 | 3.36 ± 1.63 | - 36 |
| total triglycerides (mmol/l) | 4.01 ± 2.14 | 3.50 ± 2.61 | - 36 | 3.39± 0.86 | 3.27 ± 2.78 | - 3 | 1.11 ± 0.31 | 1.38 ± 0.46 | + 25 |
| VLDL cholesterol (mmol/l) | 5.38 ± 4.53 | 3.0 ± 2.84 | - 44 | 2.35 ± 0.78 | 1.53 ± 1.50 | - 35 | 0.34 ± 0.31 | 0.41 ± 0.28 | + 19 |
| VLDL triglycerides (mmol/l) | 3.85 ± 2.14 | 3.29 ± 2.58 | - 14 | 3.21 ± 0.73 | 3.14 ± 2.72 | - 2 | 0.82 ± 0.38 | 1.19 ± 0.56 | + 45 |
| Apolipoprotein B (mg/dl) | 91 ± 32 | 52 ± 11 | - 42 | 144 ± 34 | 70 ± 25 | - 52 | 141 ± 49 | 85 ± 33 | - 40 |
| Lipoprotein(a) (mg/dl) | 44 ± 31 | 35 ± 29 | - 20 | 44 ± 79 | 26 ± 47 | - 41 | 67 ± 54 | 48 ± 40 | - 28 |

[a] Refers to % change observed during therapy with PCSK9 inhibitors.

The effect on lipoprotein subfractions (IDL-HDL) is shown in Fig 2 (cholesterol) and Fig 3 (apoB). In hypercholesterolemia and mixed hyperlipidemia, the fractions containing most of apoB and cholesterol are fractions 5–11, representing LDL, while in FDBL, cholesterol and apoB are more equally distributed between fractions 1–11, which contain IDL and LDL. PCSK9 inhibition reduces fractions 5–11 in hypercholesterolemia and 1–11 in mixed hyperlipidemia and FDBL.

## Discussion

PCSK9 inhibition reduces the plasma concentration of all apoB-containing lipoproteins except large, triglyceride-rich VLDL. This is true in patients with isolated LDL-hypercholesterolemia, with mixed hyperlipidemia and with familial dysbetalipoproteinemia. This observation is consistent with a dramatically increased LDL-receptor activity resulting in the uptake of all apoB-containing lipoproteins, except when the apoB receptor binding region is not accessible, as is the case for large triglyceride-rich VLDL [16].

It is noteworthy that in both hypertriglyceridemic patient groups (familial dysbetalipoproteinemia and mixed hyperlipidemia) VLDL-cholesterol is decreased to a larger extent than

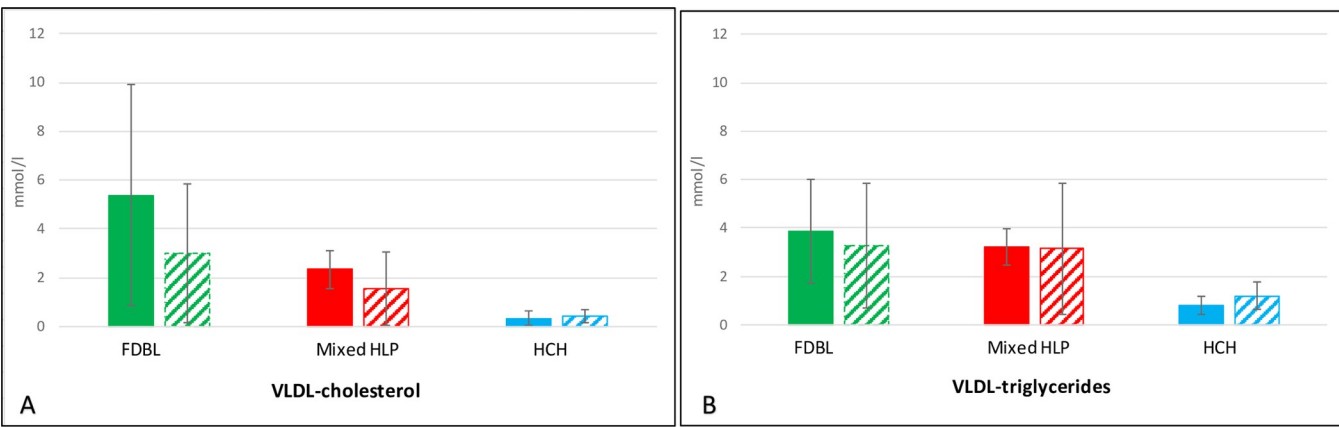

**Fig 1. VLDL-cholesterol (panel A) and VLDL-triglycerides (panel B) before (baseline, filled columns) and after 12 weeks (striped columns) of PCSK9 inhibition.** Different colors indicate different patient groups. Shown are means ± SD; green: FDBL, familial dysbetalipoproteinemia; red: mixed HLP, mixed hyperlipidemia; blue: HCH, hypercholesterolemia.

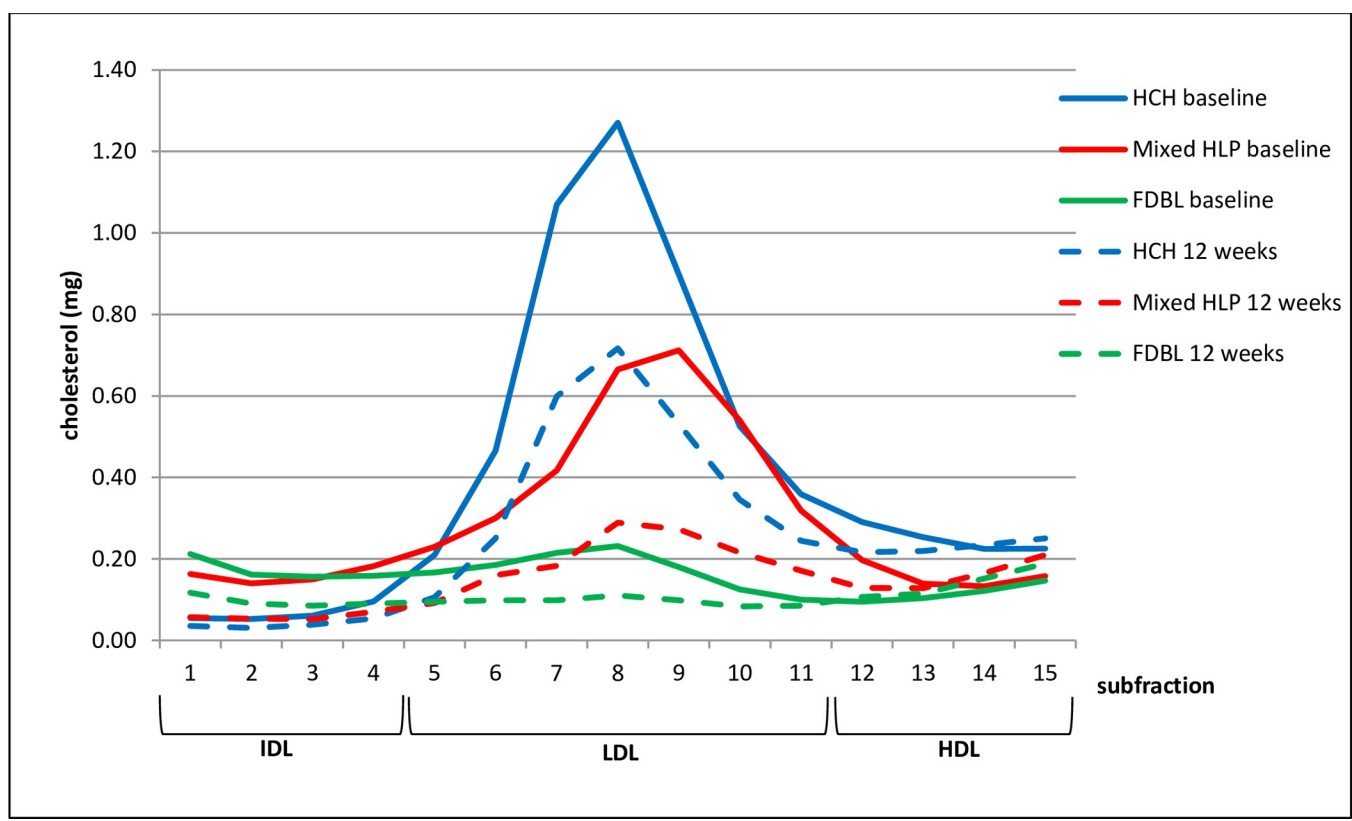

**Fig 2. Cholesterol content (means) in different lipoprotein subfractions before (baseline, solid line) and after 12 weeks (dashed line) of PCSK9 inhibition.** Different colors indicate different patient groups. HCH, hypercholesterolemia; mixed HLP, mixed hyperlipidemia; FDBL, familial dysbetalipoproteinemia.

VLDL-triglycerides, indicating that there might be distinct groups of particles, some of which can be reduced by an increased LDL-receptor activity while others cannot be reduced. In absolute terms, VLDL-cholesterol is much higher at baseline in FDBL than in mixed hyperlipidemia. This is consistent with the known lipoprotein abnormalities in FDBL, where mostly the concentration of remnant particles is increased. In mixed hyperlipidemia (not FDBL) presumably fewer lipoproteins floating in the VLDL-fraction are removed by an increased LDL-receptor activity, because most of the triglyceride-rich lipoproteins observed in mixed hyperlipidemia are "typical" VLDL with little affinity to the LDL-receptor.

It can be assumed that the "anti-atherosclerotic" potential of PCSK9-inhibition is similar in FDBL, mixed hyperlipidemia and hypercholesterolemia as the concentration of apoB-containing lipoproteins is reduced to a similar extent. While in patients with mixed hyperlipidemia and hypercholesterolemia the reduction of apoB-containing lipoproteins occurs primarily in the LDL-fraction, a much broader range of particles, though mostly VLDL, is reduced in FDBL. This is confirmed by the effect of PCSK9-inhibition on lipoprotein subfractions (Figs 2 and 3). Cholesterol and apoB are reduced over the whole range of apoB-containing lipoproteins in FDBL. Although all fractions are reduced in proportion, in absolute terms, the most prominent fraction is reduced the most. The reduction of apoB containing lipoproteins also extends to lipoprotein(a), which was reduced in all 3 patient groups. It can be assumed that this contributes to the risk reduction, although the relative importance is still controversial [17].

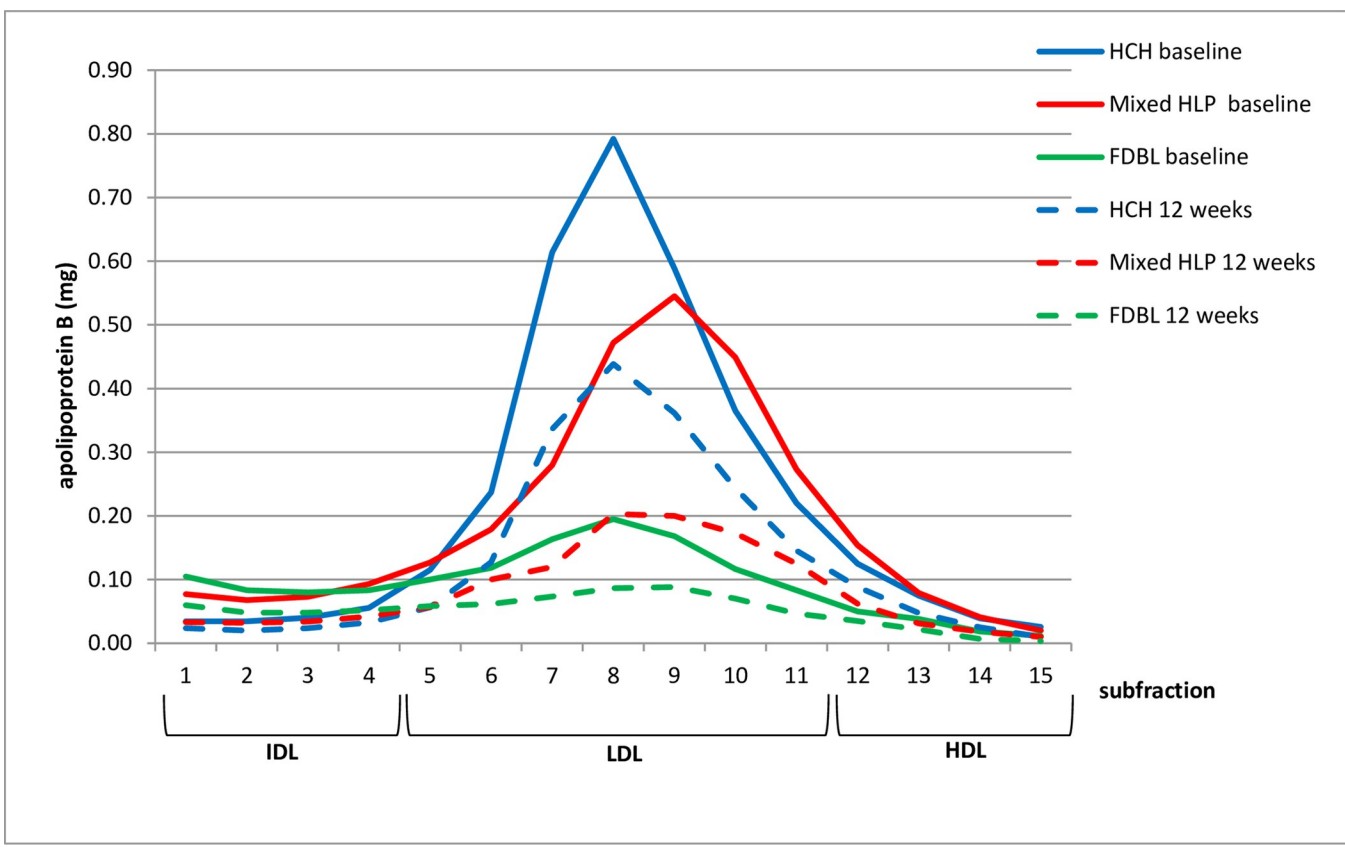

**Fig 3. ApoB content (means) in different lipoprotein subfractions before (baseline, solid line) and after 12 weeks (dashed line) of PCSK9 inhibition.** Different colors indicate different patient groups. HCH, hypercholesterolemia; mixed HLP, mixed hyperlipidemia; FDBL, familial dysbetalipoproteinemia.

The effect of statins, fibrates, ezetimibe and PCSK9-inhibititors on lipoprotein subfractions has been studied in the past, with no data (to our knowledge) in patients with FDBL [18–31]. Statins predominantly decrease intermediate dense LDL with some discordant results (some showing similar reductions in larger and smaller LDL while others show a predominant effect on intermediate dense LDL). Fibrates generally induce a shift in subfraction distribution (from small dense LDL towards less dense LDL) [24, 29]. A reduction of all LDL-subfraction was also seen during PCSK9-inhibititor treatment [30, 31].

Theoretically, the data are also consistent with a model in which PCSK9-inhibition primarily induces an elevated uptake of intermediate dense LDL particles and at the same time enhances the delipidation of larger lipoproteins to LDL particles. However, from a physiological point of view it is more likely that the increased activity of the LDL-receptor results in a direct uptake of a wide range of apoB-containing lipoproteins (whichever is available, as long as the receptor-binding region is accessible). The effect of PCSK9 inhibition on triglycerides has been discussed in the past, generally showing little effect on plasma triglyceride levels. But more recent metabolic studies indicate that PCSK9 inhibitors may enhance the clearance of triglyceride rich postprandial lipoproteins in obesity [32, 33]. This is consistent with the data shown here.

A limitation of the study represents the small number of subjects evaluated. Thus, we could not study the wide range of lipid abnormalities (normal lipid status, hypertriglyceridemia, hypercholesterolemia, mixed hyperlipidemia) shown in FDBL patients. In our study, we evaluated patients with the typical dyslipidemia and therefore our conclusions are also limited to

this patient subgroup. On the other hand, it is very likely that a similar result can also be expected in patients with FDBL presenting with other lipid phenotypes. Another limitation is that we only studied the effect of evolocumab, but again it is likely that results would be identical with other PCSK9-targeting therapies.

Taken together, our results indicate that PCSK9-inhibition reduces the concentration of a wide variety of apoB-containing lipoproteins with the dominant particle size being reduced the most. Thus, in LDL hypercholesterolemia or mixed hyperlipidemia where most apoB-containing lipoproteins are LDL, the LDL fraction is reduced while the concentration of a broader range of lipoproteins is reduced in FDBL. It can be safely assumed that this reduction in apoB-containing lipoproteins translates into clinical benefit.

## Author Contributions

**Conceptualization:** Elisa Waldmann, Klaus G. Parhofer.

**Data curation:** Elisa Waldmann, Liya Wu, Kristina Busygina, Julia Altenhofer, Kerstin Henze, Klaus G. Parhofer.

**Formal analysis:** Elisa Waldmann, Klaus G. Parhofer.

**Investigation:** Elisa Waldmann, Liya Wu, Kristina Busygina, Julia Altenhofer, Kerstin Henze, Alexander Folwaczny.

**Methodology:** Elisa Waldmann, Julia Altenhofer, Kerstin Henze, Klaus G. Parhofer.

**Resources:** Klaus G. Parhofer.

**Supervision:** Klaus G. Parhofer.

**Validation:** Elisa Waldmann, Alexander Folwaczny, Klaus G. Parhofer.

**Writing – original draft:** Klaus G. Parhofer.

**Writing – review & editing:** Elisa Waldmann, Liya Wu, Kristina Busygina, Julia Altenhofer, Alexander Folwaczny, Klaus G. Parhofer.

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
