## [Decision Letter · Decision Letter 0]

29 Nov 2021

PONE-D-21-26920

Effect of PCSK9 inhibition on lipoprotein subfractions in familial dysbetalipoproteinemia (type III hyperlipidemia)

PLOS ONE

Dear Dr. Parhofer,

Thank you for submitting your manuscript to PLOS ONE. After careful consideration, we feel that it has merit but does not meet PLOS ONE’s publication criteria as it currently stands. Therefore, we invite you to submit a revised version of the manuscript that addresses all the points raised by the Reviewers..

We look forward to receiving your revised manuscript.

Kind regards,

Laura Calabresi

Academic Editor

PLOS ONE

“I have read the journal's policy and the authors of this manuscript have the following competing interests: KGP has received research funding and/or honoraria for consultancy and/or speaker’s bureau and/or DMC activity from:

Akcea, Amarin, Amgen, Berlin-Chemie, Biomarin, Boehringer-Ingelheim, Dr. Schär, Daiichi-Sankyo, MSD, Novartis, Regeneron, Sanofi, and Silence Therapeutics”

Reviewers' comments:

Reviewer's Responses to Questions

**Comments to the Author**

1. Is the manuscript technically sound, and do the data support the conclusions?

Reviewer #1: Yes

Reviewer #2: Partly

Reviewer #3: Partly

2. Has the statistical analysis been performed appropriately and rigorously? 

Reviewer #1: I Don't Know

Reviewer #2: N/A

Reviewer #3: No

3. Have the authors made all data underlying the findings in their manuscript fully available?

Reviewer #1: Yes

Reviewer #2: Yes

Reviewer #3: No

4. Is the manuscript presented in an intelligible fashion and written in standard English?

Reviewer #1: Yes

Reviewer #2: Yes

Reviewer #3: Yes

5. Review Comments to the Author

Reviewer #1: Although I believe the study is interesting, it contains some issues that need to be sorted out.

The introduction misses the part relative to PCSK9, e.g., the role of PCSK9 on the expression of LDL-R, and the general impact of PCSK9 inhibition in terms of LDL-C lowering and atherosclerotic cardiovascualr disease (ASCVD) risk reduction. Concerning the biology of PCSK9, the Authors can refer to a recent review (PMID: 34019847). A similar comment is valid in the case of lipoprotein(a). Relative to this, the discussion lacks to mention the impact of PCSK9 inhibition on lipoprotein(a). Moreover, lipoprotein(a) levels seem to be very high and in the hypercholesterolemic group, they overtake the ASCVD risk threshold of 50 mg/dL. The Authors should evaluate the contribution of cholesterol carried by lipoprotein(a), and calculate, in addition, the corrected LDL-C (PMID: 34450317). Considering that, for lipoprotein(a), conversion between mg/dL and mmol/L is not advisable, cholesterol should be expressed both in mmol/L and mg/dL. Please report the assay that has been used to measure lipoprotein(a).

Relative to ultracentrifugation, is it correct “48 hours at 158°C?”

Which statistical test has been used to evaluate differneces? My apology if I missed it.

The impact of PCSK9 on triglycerides should be discussed in more details. Please refer to the review by Dijk W et al (PMID: 29665987) or Baragetti et al (PMID: 29428206).

Which is the percentage of recurrence of FDBL? This could be reported both in the introduction and in the limitation section. Please report that it is a single centre study. The title should report that data have been obtained with evolocumab.

Reviewer #2: In this study, the Authors investigated the impact of PCSK9 inhibition on lipoproteins in patients with FDBL. A comparison with patients with LDL-hypercholesterolemia and mixed hyperlipidemia has been made. PCSK9 inhibitors reduced cholesterol and apoB in all patients. In FDBL they specifically reduced VLDL-cholesterol. The Authors concluded that PCSK9-inhibition reduces the concentration of a wide variety of apoB-containing lipoproteins with the dominant particle size being reduced the most.

The study is original and interesting, however the paper is only descriptive, since sample size is too small and statistical analysis could not be performed.

This reviewer is aware of the rareness of familial dysbetalipoproteinemia, however three patients are not so much to draw robust conclusions, mostly because of the huge variability in lipid and lipoprotein profile.

Thus, more efforts should be dedicated to enroll additional subjects and increase sample size.

Other comments:

Tables: how are data reported? Please provide whether the number are mean ± SD or median ± SEM.

Table 1: which is the prevalence of metabolic syndrome in these patients.

Figure 2 is confusing. I would suggest using 3 different panels.

Non si reduce solo il picco più alto. Si riduce tutto in proporzione

State which inhibitor has been used also in method section, since it is reported only in the abstract.

Reviewer #3: The study is interesting but some consideration has to be done.

Please consider revising the description of the disease.

Aim. From your introduction, I do not understand what you want to test by comparing these population. Which differences do you expect? Why this study is relevant?

LDL-hypercholesterolemia, I imagine that these patients are heterozygous FH. Please use the correct terminology.

I do not understand the population selection:

1) Please do not give in methods any information on the demographic or clinical characteristics of these patients. These are reported in Table 1.

2)Did you genotype your patients? Both ApoE2/E2 patients and HeFH have already received a molecular confirmation? If yes, was it done for scientific purpose for this study or for clinical reasons?

3)Nevertheless, how did you choose patients with mixed dyslipidemia and HeFH to be compared with FDBL. Did you take all your patients taking PCKS9i?

“All had the indication for treatment with PCSK9-inhibitors according to German

regulatory standards (not reaching treatment goals recommended by the European guidelines despite

maximally tolerated oral lipid-lowering therapy)” please give a reference otherwise it is difficult to understand patient’s characteristics.

Where PCKS9i given in a study protocol? I do not understand the study design. I guess if it is an observational real-world study or an interventional trial.

As you assess only lipoprotein fraction at baseline and 3 months a suggest you consider in your discussion that your results might be in “acute” phase of treatment.

Add significance to Table 1 (BMI is significantly different between groups)

Please define CVD, CAD and PVD. Did you consider only acute events or all the events?

Please also give statistics and the methodology used to analyze the data (reporting it in a separate chapter.

Inferential statistics is completely lacking. You should add it throughout the manuscript (including Table 2 and figures)

Moreover, there was any changes in the background lipid lowering therapies in the 3 months of treatment? Please add lipid lowering therapies to table 2.

6. PLOS authors have the option to publish the peer review history of their article (what does this mean?). If published, this will include your full peer review and any attached files.

Reviewer #1: No

Reviewer #2: No

Reviewer #3: No

---

## [Author Response · Author response to Decision Letter 0]

21 Dec 2021

We thank the Editor and the reviewers for their thoughtful and constructive comments. We have incorporated them into the revised version and believe that this has considerably improved the manuscript. For details look at the upöoaded documents (Response to Reviewers; Cover letter R1). We hope that the manuscript will now be fond acceptable for publication.

---

## [Decision Letter · Decision Letter 1]

27 Jan 2022

PONE-D-21-26920R1Effect of PCSK9 inhibition with evolocumab on lipoprotein subfractions in familial dysbetalipoproteinemia (type III hyperlipidemia)PLOS ONE

Dear Dr. Parhofer,

Thank you for submitting your manuscript to PLOS ONE. After careful consideration, we feel that it has merit but does not fully meet PLOS ONE’s publication criteria as it currently stands. Therefore, we invite you to submit a revised version of the manuscript that addresses the points raised during the review process.

Authors must solve issues raised by Reviewer #3, who I agree with.

We look forward to receiving your revised manuscript.

Kind regards,

Laura Calabresi

Academic Editor

PLOS ONE

Reviewers' comments:

Reviewer's Responses to Questions

**Comments to the Author**

1. If the authors have adequately addressed your comments raised in a previous round of review and you feel that this manuscript is now acceptable for publication, you may indicate that here to bypass the “Comments to the Author” section, enter your conflict of interest statement in the “Confidential to Editor” section, and submit your "Accept" recommendation.

Reviewer #1: (No Response)

Reviewer #2: All comments have been addressed

Reviewer #3: (No Response)

2. Is the manuscript technically sound, and do the data support the conclusions?

Reviewer #1: (No Response)

Reviewer #2: Partly

Reviewer #3: Partly

3. Has the statistical analysis been performed appropriately and rigorously? 

Reviewer #1: (No Response)

Reviewer #2: N/A

Reviewer #3: N/A

4. Have the authors made all data underlying the findings in their manuscript fully available?

Reviewer #1: (No Response)

Reviewer #2: (No Response)

Reviewer #3: Yes

5. Is the manuscript presented in an intelligible fashion and written in standard English?

Reviewer #1: (No Response)

Reviewer #2: Yes

Reviewer #3: Yes

6. Review Comments to the Author

Reviewer #1: (No Response)

Reviewer #2: Almost all comments have been addressed. Sample size remains a major issue that cannot be resolved.

Minor comments:

Please revise abbreviations and typos (e.g. Celcius, PCSK-9 or PCSK9)

Reviewer #3: Dear author my major concearn is the lack of an accurate definition of the patients included in the study.

In fact, I am awared that familial dysbetalipoproteinemia is rare but for this reason, patients should be very well characterized. Similarly, the "control" population lack of definition as for example, you have included patients with FH with those suffering with other form of hypercholesterolemia in a unique group. I believe that these could be potentially confounders. I suggest publishing this data as a short communication rather than as an original article.

7. PLOS authors have the option to publish the peer review history of their article (what does this mean?). If published, this will include your full peer review and any attached files.

Reviewer #1: No

Reviewer #2: No

Reviewer #3: No

---

## [Author Response · Author response to Decision Letter 1]

7 Feb 2022

Comments were addressed in the revised version of the manuscript. Please see "Cover letter" and "Response to reviewers".

Regards

Klaus Parhofer

---

## [Decision Letter · Decision Letter 2]

9 Mar 2022

Effect of PCSK9 inhibition with evolocumab on lipoprotein subfractions in familial dysbetalipoproteinemia (type III hyperlipidemia)

PONE-D-21-26920R2

Dear Dr. Parhofer,

We’re pleased to inform you that your manuscript has been judged scientifically suitable for publication and will be formally accepted for publication once it meets all outstanding technical requirements.

Kind regards,

Laura Calabresi

Academic Editor

PLOS ONE

Additional Editor Comments (optional):

Reviewers' comments:

Reviewer's Responses to Questions

**Comments to the Author**

1. If the authors have adequately addressed your comments raised in a previous round of review and you feel that this manuscript is now acceptable for publication, you may indicate that here to bypass the “Comments to the Author” section, enter your conflict of interest statement in the “Confidential to Editor” section, and submit your "Accept" recommendation.

Reviewer #2: All comments have been addressed

Reviewer #3: All comments have been addressed

2. Is the manuscript technically sound, and do the data support the conclusions?

Reviewer #2: Partly

Reviewer #3: Yes

3. Has the statistical analysis been performed appropriately and rigorously? 

Reviewer #2: N/A

Reviewer #3: Yes

4. Have the authors made all data underlying the findings in their manuscript fully available?

Reviewer #2: Yes

Reviewer #3: Yes

5. Is the manuscript presented in an intelligible fashion and written in standard English?

Reviewer #2: Yes

Reviewer #3: Yes

6. Review Comments to the Author

Reviewer #2: (No Response)

Reviewer #3: Thank you for your replay and having added further information on patients. I still believe that molecular diagnosis could have added some further interesting to this observation.

7. PLOS authors have the option to publish the peer review history of their article (what does this mean?). If published, this will include your full peer review and any attached files.

Reviewer #2: No

Reviewer #3: No

---

## [Editor Report · Acceptance letter]

15 Mar 2022

PONE-D-21-26920R2 

Effect of PCSK9 inhibition with evolocumab on lipoprotein subfractions in familial dysbetalipoproteinemia (type III hyperlipidemia) 

Dear Dr. Parhofer:

I'm pleased to inform you that your manuscript has been deemed suitable for publication in PLOS ONE. Congratulations! Your manuscript is now with our production department. 

Kind regards, 

on behalf of

Prof. Laura Calabresi 

Academic Editor

PLOS ONE